# On the Sample Complexity of Stabilizing LTI Systems on a Single Trajectory

**Yang Hu**
SEAS, Harvard University
Massachusetts, USA
yanghu@g.harvard.edu

**Adam Wierman**
CMS, California Institute of Technology
California, USA
adamw@caltech.edu

**Guannan Qu**
ECE, Carnegie Mellon University
Pennsylvania, USA
gqu@andrew.cmu.edu

## Abstract

Stabilizing an unknown dynamical system is one of the central problems in control theory. In this paper, we study the sample complexity of the learn-to-stabilize problem in Linear Time-Invariant (LTI) systems on a single trajectory. Current state-of-the-art approaches require a sample complexity linear in $n$, the state dimension, which incurs a state norm that blows up exponentially in $n$. We propose a novel algorithm based on spectral decomposition that only needs to learn "a small part" of the dynamical matrix acting on its unstable subspace. We show that, under proper assumptions, our algorithm stabilizes an LTI system on a single trajectory with $O(k \log n)$ samples, where $k$ is the instability index of the system. This represents the first sub-linear sample complexity result for the stabilization of LTI systems under the regime when $k = o(n)$.

## 1 Introduction

Linear Time-Invariant (LTI) systems, namely $x_{t+1} = Ax_t + Bu_t$, where $x_t \in \mathbb{R}^n$ is the state and $u_t \in \mathbb{R}^m$ is the control input, are one of the most fundamental dynamical systems in control theory, and have wide applications across engineering, economics and societal domains. For systems with known dynamical matrices $(A, B)$, there is a well-developed theory for designing feedback controllers with guaranteed stability, robustness, and performance [1, 2]. However, these tools cannot be directly applied when $(A, B)$ is unknown.

Driven by the success of machine learning [3, 4], there has been significant interest in learning-based (adaptive) control, where the learner does not know the underlying system dynamics and learns to control the system in an online manner, usually with the goal of achieving low regret [5–13].

Despite the progress, an important limitation in this line of work is a common assumption that the learner has a priori access to a known *stabilizing* controller. This assumption simplifies the learning task, since it ensures a bounded state trajectory in the learning stage, and thus enables the learner to learn with low regret. However, assuming a known stabilizing controller is not practical, as *stabilization* itself is nontrivial and considered equally important as any other performance guarantee.

To overcome this limitation, in this paper we consider the *learn-to-stabilize* problem, i.e., learning to stabilize an unknown dynamical system without prior knowledge of any stabilizing controller.

†This work is supported by NSF Grants CNS-2146814, CPS-2136197, CNS-2106403, NGSDI-2105648, EPCN-2154171, with additional support from Amazon AWS.

36th Conference on Neural Information Processing Systems (NeurIPS 2022).

Understanding the learn-to-stabilize problem is of great importance to the learning-based control literature, as it serves as a precursor to any learning-based control algorithms that assume knowledge of a stabilizing controller.

The learn-to-stabilize problem has attracted extensive attention recently. For example, [14] and [15] adopt a model-based approach that first excites the open-loop system to learn dynamical matrices $(A, B)$, and then designs a stabilizing controller, with a sample complexity scaling linearly in $n$, the state dimension. However, a linearly-scaling sample complexity could be unsatisfactory for some specific instances, since the state trajectory still blows up exponentially when the open-loop system is unstable, incurring a $2^{\tilde{\Theta}(n)}$ state norm, and hence a $2^{\tilde{\Theta}(n)}$ regret (in LQR settings, for example). Another recent work [16] proposes a policy-gradient-based discount annealing method that solves a series of discounted LQR problems with increasing discount factors, and shows that the control policy converges to a near-optimal policy. However, this model-free approach only guarantees a $\mathrm{poly}(n)$ sample complexity. In fact, to the best of our knowledge, state-of-the-art learn-to-stabilize algorithms with theoretical guarantees always incur state norms exponential in $n$.

It has been shown in [15] that all *general-purpose* control algorithms are doomed to suffer a *worst-case* regret of $2^{\Omega(n)}$. This result is intuitive, since from an information-theoretic perspective, a complete recovery of $A$ should take $\Theta(n)$ samples since $A$ itself involves $n^2$ parameters. However, this does not rule out the possibility that we can achieve better regret in *specific* systems. Our work is motivated by the observation that it is not always necessary to learn the whole matrix $A$ to stabilize an LTI system. For example, if the system is open-loop stable, we do not need to learn anything to stabilize it. For general LTI systems, it is still intuitive that open-loop *stable "modes"* exist and need not be learned for the learn-to-stabilize problem. So, we focus on learning a controller that stabilizes only the *unstable "modes"*, making it possible to learn a stabilizing controller without exponentially exploding state norms. The central question of this paper is:

> *Can we exploit instance-specific properties of an LTI system to learn to stabilize it*
> *on a single trajectory, without incurring a state norm exponentially large in $n$?*

**Contribution.** In this paper, we answer the above question by designing an algorithm that stabilizes an LTI system with only $O(k \log n)$ state samples along a single trajectory, where $k$ is the *instability index* of the open-loop system and is defined as the number of unstable "modes" (i.e., eigenvalues with moduli larger than 1) of matrix $A$. Our result is significant in the sense that $k$ can be considerably smaller than $n$ for practical systems and, in such cases, our algorithm stabilizes the system using asymptotically fewer samples than prior work; specifically, it only incurs a state norm (and regret) in the order of $2^{O(k \log n)}$, much smaller than $2^{O(n)}$ of prior state of the art when $k \ll n$.

To formalize the concept of unstable "modes" for the presentation of our algorithm and analysis, we formulate a novel framework based on the spectral decomposition of dynamical matrix $A$. More specifically, we focus on the *unstable subspace* $E_\mathrm{u}$ spanned by the eigenvectors corresponding to unstable eigenvalues, and consider the system dynamics "restricted" to it — states are orthogonally projected onto $E_\mathrm{u}$, and we only have to learn the effective part of $A$ within subspace $E_\mathrm{u}$, which takes only $O(k)$ samples. The formulation is explained in detail in Section 3.1 and Appendix A. We comment that this idea of decomposition is in stark contrast to prior work, which in one way or another seeks to learn the entire $A$ (or other similar quantities).

**Related work.** Our work contributes to and builds upon related works described below.

*Learning for control assuming known stabilizing controllers.* There has been a large literature on learning-based control with known stabilizing controllers. For example, one line of research utilizes model-free policy optimization approaches to learn the optimal controller for LTI systems [5–7, 17–30]. All of these works require a known stabilizing controller as an initializer for the policy search method. Another line of research uses model-based methods, i.e., learning dynamical matrices $(A, B)$ first before designing a controller, which also require a known stabilizing controller (e.g., [31–39]). Compared to these works, we focus on the learn-to-stabilize problem without knowledge of an initial stabilizing controller, which can serve as a precursor to existing learning-for-control works that require a known stabilizing controller.

*Learning to stabilize on a single trajectory.* Stabilizing linear systems over *infinite* horizons with asymptotic convergence guarantees is a classical problem that has been studied extensively in a wide range of papers such as [40–42]. On the other hand, the problem of system stabilization over *finite* horizons remains partially open and has not seen significant progresses. Algorithms incurring

a $2^{O(n)}O(\sqrt{T})$ regret have been proposed in settings that rely on relatively strong assumptions of controllability and strictly stable transition matrices [13, 43], which has recently been improved to $2^{\tilde{O}(n)} + \tilde{O}(\text{poly}(n)\sqrt{T})$ [14, 15]. Another model-based approach that merely assumes stabilizability is introduced in [44], though it does not provide guarantees on regret or sample complexity. A more recent model-free approach based on policy gradient [16] provides a novel perspective into this problem, yet it can only guarantee a $\text{poly}(n)$ sample complexity. Compared to these previous works, our approach requires only $O(k \log n)$ samples, incurring a sub-exponential state norm.

Another recent work [45] proposes to do partial system identification via projecting the state onto a lower-dimensional subspace, which is similar in intuition with our work. However, the problem considered there is system stabilization with a fixed initial state $x_0$, and their approach only eliminates the unstable component along that specific trajectory in $k$ steps when $x_0$ lies in a $k$-dimensional subspace. In contrast, our approach finds a stabilizing controller for the system with sub-linear sample complexity along an arbitrary trajectory regardless of the initial state.

*Learning to stabilize on multiple trajectories.* There are also works [12, 46] that do not assume known stabilizing controllers and learn the full dynamics before designing an optimal stabilizing controller. While requiring $\tilde{\Theta}(n)$ samples which is larger than $\tilde{O}(k)$ of our work, those approaches do not have the exponentially large state norm issue as they allow *multiple trajectories*; i.e., the state can be "reset" to $0$ so that it won't get too large. In contrast, we focus on the more challenging single-trajectory scenario where the state cannot be reset.

*System Identification.* Our work is also related to the system identification literature, which focuses on learning the system parameters of dynamical systems, with early works like [47] focusing on asymptotic guarantees, and more recent works such as [48–53] focusing on finite-time guarantees. Our approach also identifies the system (partially) before constructing a stabilizing controller, but we only identify a part of $A$ rather than the entire $A$.

## 2  Problem Formulation

We consider a noiseless LTI system $x_{t+1} = Ax_t + Bu_t$, where $x_t \in \mathbb{R}^n$ and $u_t \in \mathbb{R}^m$ are the *state* and *control input* at time step $t$, respectively. The dynamical matrices $A \in \mathbb{R}^{n \times n}$ and $B \in \mathbb{R}^{n \times m}$ are unknown to the learner. The learner is allowed to learn about the system by interacting with it on a *single trajectory* — the initial state is sampled uniformly at random from the unit hyper-sphere surface in $\mathbb{R}^n$, and then, at each time step $t$, the learner is allowed to observe $x_t$ and freely determine $u_t$. The goal of the learner is to learn a stabilizing controller, which is defined as follows.

**Definition 2.1** (Stabilizing Controller). *Control rule $u_t = f_t(x_t, x_{t-1}, \cdots, x_0)$ is called a **stabilizing controller** if and only if the closed-loop system $x_{t+1} = Ax_t + Bu_t$ is asymptotically stable; i.e., for any $x_0 \in \mathbb{R}^n$, $\lim_{t \to \infty} \|x_t\| = 0$ is guaranteed in the closed-loop system.*

To achieve this goal, a simple strategy is to let the system run in open loop to learn $(A, B)$ via least squares, and then design a stabilizing controller based on the learned dynamical matrices. However, as has been discussed in the introduction, such a simple strategy inevitably induces an exponentially large stage norm that is potentially improvable.[1] A possible remedy for this is to learn "a small part" of $(A, B)$ that is crucial for stabilization. Driven by such intuition, the core problem of this paper is to characterize what is the "small part" and design an algorithm to learn it.

Note that, although it is common to include an additive disturbance term $w_t$ in the LTI dynamics, the introduction of stochasticity does not provide additional insights into our decomposition-based algorithm, but rather, merely makes the analysis more technically challenging. Therefore, here we simply omit the noise in theoretical results for the clarity of exposition, and will show by numerical experiments that our algorithm can also handle disturbances (see Appendix H).

**Notation.** For $z \in \mathbb{C}$, $|z|$ is the modulus of $z$. For a matrix $A \in \mathbb{R}^{p \times q}$, $A^\top$ denotes the transpose of $A$; $\|A\|$ is the induced 2-norm of $A$ (equal to its largest singular value), and $\sigma_{\min}(A)$ is the smallest singular value of $A$; when $A$ is square, $\rho(A)$ denotes the spectral radius of $A$, and $\kappa_e(A)$ denotes the condition number of the matrix consisting of $A$'s eigenvectors as columns. The space spanned by

---

[1]More sophisticated exploration strategies might be adopted to learn $(A, B)$ [13, 15, 44], but as long as the control inputs do not completely cancel out the "dominant part" of the states, the above intuition still holds to a large extent as the 'dominant part" of the state is still blowing up exponentially.

$\{v_1, \cdots, v_p\}$ is denoted by $\mathrm{span}(v_1, \cdots, v_p)$, and the column space of $A$ is denoted by $\mathrm{col}(A)$. For two subspaces $U, V$ of $\mathbb{R}^n$, $U^\perp$ is the orthogonal complement of $U$, and $U \oplus V$ is the direct sum of $U$ and $V$. The zero matrix and identity matrix are denoted by $\mathbf{0}$, $I$, respectively.

## 3  Learning to Stabilize from Zero (LTS$_0$)

The core of this paper is a novel algorithm, Learning to Stabilize from Zero (LTS$_0$), that utilizes a decomposition of the state space based on a characterization of the notion of unstable "modes". The decomposition and other preliminaries for the algorithm are first introduced in Section 3.1, and then we proceed to describe LTS$_0$ in Section 3.2.

### 3.1  Algorithm Preliminaries

We first introduce the decomposition of the state space in Section 3.1.1, which formally defines the "small part" of $A$ mentioned in the introduction. Then, we introduce $\tau$-hop control in Section 3.1.2, so that we can construct a stabilizing controller based only on the "small part" of $A$ (as opposed to the entire $A$). Together, these two ideas form the basis of LTS$_0$.

#### 3.1.1  Decomposition of the State Space

Consider the open-loop system $x_{t+1} = Ax_t$. Suppose that $A$ is diagonalizable, and let $\lambda_1, \cdots, \lambda_n$ denote the eigenvalues of $A$, which are assumed to be distinct and satisfy

$$|\lambda_1| \geq |\lambda_2| \geq \cdots \geq |\lambda_k| > 1 > |\lambda_{k+1}| \geq \cdots \geq |\lambda_n|.$$

We define the *eigenspaces* associated to these eigenvalues: for a real eigenvalue $\lambda_i \in \mathbb{R}$ corresponding to eigenvector $v_i \in \mathbb{R}^n$, we associate with it a 1-dimensional space $E_i = \mathrm{span}(v_i)$; for a complex eigenvalue $\lambda_i \in \mathbb{C} \setminus \mathbb{R}$ corresponding to eigenvector $v_i \in \mathbb{C}^n$, there must exist some $j$ such that $\lambda_j = \bar{\lambda}_i$ (corresponding to eigenvector $v_j = \bar{v}_i$), and we associate with them a 2-dimensional space $E_i = E_j = \mathrm{span}((v_i + \bar{v}_i), \mathrm{i}(v_i - \bar{v}_i))$. Further, define the *unstable subspace* $E_\mathrm{u} := \bigoplus_{i \leq k} E_i$ and *stable subspace* $E_\mathrm{s} := \bigoplus_{i > k} E_i$.

As discussed earlier, we only need to learn "a small effective part" of $A$ associated with the unstable "modes", or the unstable eigenvectors of $A$. For this purpose, in the following we formally define a decomposition based on the orthogonal projection onto the unstable subspace $E_\mathrm{u}$. This decomposition forms the foundation of our algorithm.

**The $E_\mathrm{u} \oplus E_\mathrm{u}^\perp$-decomposition.** Suppose the unstable subspace $E_\mathrm{u}$ and its orthogonal complement $E_\mathrm{u}^\perp$ are given by *orthonormal* bases $P_1 \in \mathbb{R}^{n \times k}$ and $P_2 \in \mathbb{R}^{n \times (n-k)}$, respectively, namely

$$E_\mathrm{u} = \mathrm{col}(P_1), \ E_\mathrm{u}^\perp = \mathrm{col}(P_2).$$

Let $P = [P_1 \ P_2]$, which is also orthonormal and thus $P^{-1} = P^\top = [P_1 \ P_2]^\top$. For convenience, let $\Pi_1 := P_1 P_1^\top$ and $\Pi_2 = P_2 P_2^\top$ be the *orthogonal* projectors onto $E_\mathrm{u}$ and $E_\mathrm{u}^\perp$, respectively. With the state space decomposition, we proceed to decompose matrix $A$. Note that $E_\mathrm{u}$ is an invariant subspace with regard to $A$ (but $E_\mathrm{u}^\perp$ not necessarily is), there exists $M_1 \in \mathbb{R}^{k \times k}$, $\Delta \in \mathbb{R}^{k \times (n-k)}$ and $M_2 \in \mathbb{R}^{(n-k) \times (n-k)}$, such that

$$AP = P \begin{bmatrix} M_1 & \Delta \\ & M_2 \end{bmatrix} \ \Leftrightarrow \ M := \begin{bmatrix} M_1 & \Delta \\ & M_2 \end{bmatrix} = P^{-1}AP.$$

In the decomposition, the top-left block $M_1 \in \mathbb{R}^{k \times k}$ represents the action of $A$ on the unstable subspace. Matrix $M_1$, together with $P_1$, is the "small part" we discussed in the introduction. Note that $M_1$ ($P_1$) is only $k$-by-$k$ ($n$-by-$k$) and thus takes much fewer samples to learn compared to the entire $A$. It is also evident that $M_1$ inherits all unstable eigenvalues of $A$, while $M_2$ inherits all stable eigenvalues. Finally, we provide the system dynamics in the transformed coordinates. Let $y = [y_1^\top \ y_2^\top]^\top$ be the coordinate representation of $x$ in the basis formed by column vectors of $P$ (i.e., $x = Py$). The system dynamics in $y$-coordinates is

$$\begin{bmatrix} y_{1,t+1} \\ y_{2,t+1} \end{bmatrix} = P^{-1}AP \begin{bmatrix} y_{1,t} \\ y_{2,t} \end{bmatrix} + P^{-1}Bu_t = \begin{bmatrix} M_1 & \Delta \\ & M_2 \end{bmatrix} \begin{bmatrix} y_{1,t} \\ y_{2,t} \end{bmatrix} + \begin{bmatrix} P_1^\top B \\ P_2^\top B \end{bmatrix} u_t. \tag{1}$$

**The $E_\mathrm{u} \oplus E_\mathrm{s}$-decomposition.** In the above $E_\mathrm{u} \oplus E_\mathrm{u}^\perp$-decomposition, $E_\mathrm{u}^\perp$ is in general *not* an invariant subspace with respect to $A$. This can be seen from the top-right $\Delta$ block in $M$, which

represents how much of the state is "moved" by $A$ from $E_{\mathrm{u}}^{\perp}$ into $E_{\mathrm{u}}$ in one step. The absence of invariant properties in $E_{\mathrm{u}}^{\perp}$ is sometimes inconvenient in the analysis. Hence, we introduce another invariant decomposition that is used in the proof as follows. Specifically, $\mathbb{R}^n$ can be naturally decomposed into $E_{\mathrm{u}} \oplus E_{\mathrm{s}}$, and further both $E_{\mathrm{u}}$ and $E_{\mathrm{s}}$ are invariant with respect to $A$. We also represent $E_{\mathrm{u}} = \mathrm{col}(Q_1)$ and $E_{\mathrm{s}} = \mathrm{col}(Q_2)$ by their *orthonormal* bases, and define $Q = [Q_1\, Q_2]$. Note that, these two subspaces are generally not orthogonal, so we additionally define $Q^{-1} =: [R_1^\top R_2^\top]^\top$. Details are deferred to Appendix A.1.

Lastly, we comment that when $A$ is symmetric, the $E_{\mathrm{u}} \oplus E_{\mathrm{u}}^{\perp}$- and $E_{\mathrm{u}} \oplus E_{\mathrm{s}}$-decompositions are identical because $E_{\mathrm{u}}^{\perp} = E_{\mathrm{s}}$ in such symmetric cases. While $E_{\mathrm{u}}^{\perp} \neq E_{\mathrm{s}}$ in general cases, the "closeness" between $E_{\mathrm{u}}^{\perp}$ and $E_{\mathrm{s}}$ also contributes to the sample complexity bound in Section 4. For that reason, we formally define such "closeness" between subspaces in Definition 3.1. We point out that the definition has clear geometric interpretations and leads to connections between the bases of $E_{\mathrm{s}}$ and $E_{\mathrm{u}}^{\perp}$, which is technical and thus deferred to Appendix A.2.

**Definition 3.1** ($\xi$-Close Subspaces)**.** *For $\xi \in (0, 1]$, the subspaces $E_{\mathrm{u}}^{\perp} = \mathrm{col}(P_2), E_{\mathrm{s}} = \mathrm{col}(Q_2)$ are called **$\xi$-close** to each other, if and only if $\sigma_{\min}(P_2^\top Q_2) > 1 - \xi$.*

### 3.1.2  $\tau$-hop Control

This section discusses the design of controller based only on the "small part" of $A$, i.e., the $P_1$ and $M_1$ matrices discussed in Section 3.1.1, as opposed to the entire $A$ matrix. Note that the main objective of this subsection is to introduce the idea of our controller design when $M_1$ and $P_1$ are known without errors, whereas in Section 3.2 we fully introduce Algorithm 1 that learns $M_1$ and $P_1$ before constructing the stabilizing controller.

As discussed in Section 3.1.1, we can view $M_1$ as the "restriction" of $A$ onto the unstable subspace $E_{\mathrm{u}}$ (spanned by the basis in $P_1$) and it captures all the unstable eigenvalues of $A$. Since only $M_1$ and $P_1$ are known while $M_2$ and $P_2$ are unknown, a simple idea is to "restrict" the system trajectory entirely to $E_{\mathrm{u}}$ such that the effect of $A$ is fully captured by $M_1$, the part of $A$ that is known. However, such a restriction is not possible because, even if the current state $x_t$ is in $E_{\mathrm{u}}$ (so $Ax_t$ is also in $E_{\mathrm{u}}$), $x_{t+1} = Ax_t + Bu_t$ is generally not in $E_{\mathrm{u}}$ with non-zero $u_t$. To address this issue, recall that a desirable property of the stable component is that it spontaneously dies out in open loop. Therefore, we propose the following $\tau$-*hop controller* design, where the control input is only injected every $\tau$ steps — in this way, we let the stable component die out exponentially between two consecutive control injections. Consequently, when we examine the states every $\tau$ steps, we could expect that the trajectory appears approximately "restricted to" the unstable subspace $E_{\mathrm{u}}$.

More formally, a $\tau$-hop controller only injects non-zero $u_t$ for $t = s\tau$, $s \in \mathbb{N}$. Let $\tilde{x}_s := x_{s\tau}$ and $\tilde{u}_s := u_{s\tau}$ to be the state and input every $\tau$ time steps. We can write the dynamics of the $\tau$-hop control system as $\tilde{x}_{s+1} = A^\tau \tilde{x}_s + A^{\tau-1}B\tilde{u}_s$. We also let $\tilde{y}_s$ to denote the state under $E_{\mathrm{u}} \oplus E_{\mathrm{u}}^{\perp}$-decomposition, i.e. $\tilde{y}_s = P^\top \tilde{x}_s$. Then the state evolution can be written as

$$\begin{bmatrix} \tilde{y}_{1,s+1} \\ \tilde{y}_{2,s+1} \end{bmatrix} = P^{-1}A^\tau P \begin{bmatrix} \tilde{y}_{1,s} \\ \tilde{y}_{2,s} \end{bmatrix} + P^{-1}A^{\tau-1}B\tilde{u}_s = M^\tau \begin{bmatrix} \tilde{y}_{1,s} \\ \tilde{y}_{2,s} \end{bmatrix} + \begin{bmatrix} P_1^\top A^{\tau-1}B \\ P_2^\top A^{\tau-1}B \end{bmatrix} \tilde{u}_s, \qquad (2)$$

where we define $B_\tau := P_1^\top A^{\tau-1}B$ for simplicity, and

$$M^\tau = \left( \begin{bmatrix} M_1 & \\ & M_2 \end{bmatrix} + \begin{bmatrix} \mathbf{0} & \Delta \\ & \mathbf{0} \end{bmatrix} \right)^\tau = \begin{bmatrix} M_1^\tau & \sum_{i=0}^{\tau-1} M_1^i \Delta M_2^{\tau-1-i} \\ & M_2^\tau \end{bmatrix} =: \begin{bmatrix} M_1^\tau & \Delta_\tau \\ & M_2^\tau \end{bmatrix}.$$

Now we consider a state feedback controller $\tilde{u}_s = K_1 \tilde{y}_{1,s}$ in the $\tau$-hop control system that only acts on the unstable component $\tilde{y}_{1,s}$, the closed-loop dynamics of which can then be written as

$$\tilde{y}_{s+1} = \begin{bmatrix} M_1^\tau + P_1^\top A^{\tau-1}BK_1 & \Delta_\tau \\ P_2^\top A^{\tau-1}BK_1 & M_2^\tau \end{bmatrix} \tilde{y}_s. \qquad (3)$$

In (3), the bottom-left block becomes $P_2^\top A^{\tau-1}BK_1$, which is exponentially small in $\tau$. Therefore, with a properly chosen $\tau$, the closed-loop dynamical matrix in (3) is almost block-upper-triangular with the bottom-right block very close to $\mathbf{0}$ (recall that $M_2$ is a stable matrix). As a result, if we select $K_1$ such that $M_1^\tau + P_1^\top A^{\tau-1}BK_1$ is stable, then (3) will become stable as well. There are different ways to select such $K_1$, and in this paper, we focus on the simple case that $B$ is an $n$-by-$k$ matrix and $P_1^\top A^{\tau-1}B$ is an invertible square matrix (see Assumption 4.3$'$), in which case selecting

$$K_1 = -(P_1^\top A^{\tau-1}B)^{-1}M_1^\tau \qquad (4)$$

will suffice. Note that such a controller design will also need the knowledge of $P_1^\top A^{\tau-1} B$, which has the same dimension as $M_1$ (a $k$-by-$k$ matrix) and takes only $O(k)$ additional samples to learn. For the case that $B$ is not $n$-by-$k$, similar controller design can be done (but in a slightly more involved way), and we defer the discussion to Appendix C.

We also point out that, for the case where $A$ is symmetric, selecting $\tau = 1$ should work well. This is because $\Delta_\tau = \mathbf{0}$ in (3) for the symmetric case, and therefore, the matrix in (3) will be triangular even for $\tau = 1$. This will result in a simpler algorithm and controller design, and hence a better sample complexity bound, which we will present as Theorem 4.2 in Section 4.

We end this subsection with some comments on the role of $\tau$-hop stabilizing controllers. One may wonder if the controller design proposed here would be compatible with many downstream tasks, since the closed-loop system stabilized by a $\tau$-hop controller will still experience periodical fluctuations in state norms (although in a bounded manner). However, we want to emphasize again that the $\tau$-hop controller can serve as a precursor to any online control algorithm that assumes a known stabilizing controller, which includes system identification from stable trajectories (see, e.g., [48, 50]) and controller designs using the identified system. In this way the state norm fluctuation is only transient, and does not harm to the overall performance significantly.

## 3.2 Algorithm

Our algorithm, LTS$_0$, is divided into 4 stages: (i) learn an orthonormal basis $P_1$ of the unstable subspace $E_\mathrm{u}$ (Stage 1); (ii) learn $M_1$, the restriction of $A$ onto the subspace $E_\mathrm{u}$ (Stage 2); (iii) learn $B_\tau = P_1^\top A^{\tau-1} B$ (Stage 3); and (iv) design a controller that seeks to cancel out the "unstable" $M_1$ matrix (Stage 4). This is formally described as Algorithm 1 below.

---
**Algorithm 1** LTS$_0$: Learning a $\tau$-hop Stabilizing Controller
---
1: **Stage 1: learn the unstable subspace of $A$.**
2: Run the system in open loop for $t_0$ steps for initialization.
3: Run the system in open loop for $k$ more steps and let $D \leftarrow [x_{t_0+1} \cdots x_{t_0+k}]$.
4: Calculate $\hat{\Pi}_1 \leftarrow D(D^\top D)^{-1} D^\top$.
5: Calculate the top $k$ (normalized) eigenvectors $\hat{v}_1, \cdots \hat{v}_k$ of $\hat{\Pi}_1$, and let $\hat{P}_1 \leftarrow [\hat{v}_1 \cdots \hat{v}_k]$.
6: **Stage 2: approximate $M_1$ on the unstable subspace.**
7: Solve the least squares $\hat{M}_1 \leftarrow \arg\min_{M_1 \in \mathbb{R}^{k \times k}} \mathcal{L}(M_1) := \sum_{t=t_0+1}^{t_0+k} \|\hat{P}_1^\top x_{t+1} - \hat{M}_1 \hat{P}_1^\top x_t\|^2$.
8: **Stage 3: restore $B_\tau$ for $\tau$-hop control.**
9: **for** $i \leftarrow 1, \cdots, k$ **do**
10:     Let the system run in open loop for $\omega$ time steps.
11:     Run for $\tau$ more steps with initial $u_{t_i} = \alpha \|x_{t_i}\| e_i$, where $t_i = t_0 + k + i\omega + (i-1)\tau$.
12: Let $\hat{B}_\tau \leftarrow [\hat{b}_1 \cdots \hat{b}_k]$, where the $i^\mathrm{th}$ column $\hat{b}_i \leftarrow \frac{1}{\alpha \|x_{t_i}\|}\left(\hat{P}_1^\top x_{t_i+\tau} - \hat{M}_1^\tau \hat{P}_1^\top x_{t_i}\right)$.
13: **Stage 4: construct a $\tau$-hop stabilizing controller $K$.**
14: Construct the $\tau$-hop stabilizing controller $\hat{K} \leftarrow -\hat{B}_\tau^{-1} \hat{M}_1^\tau \hat{P}_1^\top$.
---

In the remainder of this section we provide detailed descriptions of the four stages in LTS$_0$.

**Stage 1: Learn the unstable subspace of $A$.** It suffices to learn an orthonormal basis of $E_\mathrm{u}$. We notice that, when $A$ is applied recursively, it will push the state closer to $E_\mathrm{u}$. Therefore, when we let the system run in open loop (with control input $u_t \equiv 0$) for $t_0$ time steps, the ratio between the norms of unstable and stable components will be magnified exponentially, and the state lies "almost" in $E_\mathrm{u}$. As a result, the subspace spanned by the next $k$ states, i.e. the column space of $D := [x_{t_0+1} \cdots x_{t_0+k}]$, is very close to $E_\mathrm{u}$. This motivates us to use the orthogonal projector onto $\mathrm{col}(D)$, namely $\hat{\Pi}_1 = D(D^\top D)^{-1} D^\top$, as an estimation of the projector $\Pi_1 = P_1 P_1^\top$ onto $E_\mathrm{u}$. Finally, the columns of $\hat{P}_1$ are restored by taking the top $k$ eigenvectors of $\hat{\Pi}_1$ with largest eigenvalues (they should be very close to 1), which form a basis of the estimated unstable subspace.

**Stage 2: Learn $M_1$ on the unstable subspace.** Recall that $M_1$ is the "dynamical matrix" for the $E_\mathrm{u}$-component under the $E_\mathrm{u} \oplus E_\mathrm{u}^\perp$-decomposition. Therefore, to estimate $M_1$, we first calculate the coordinates of the states $x_{t_0+1:t_0+k}$ under basis $P_1$; that is, $\hat{y}_{1,t} = \hat{P}_1^\top x_t$, for $t = t_0 + 1, \ldots, t_0 + k$.

Then, we use least squares to estimate $M_1$, which minimizes the square loss over $\hat{M}_1$

$$\mathcal{L}(\hat{M}_1) := \sum_{t=t_0+1}^{t_0+k} \|\hat{y}_{1,t+1} - \hat{M}_1 \hat{y}_{1,t}\|^2 = \sum_{t=t_0+1}^{t_0+k} \|\hat{P}_1^\top x_{t+1} - \hat{M}_1 \hat{P}_1^\top x_t\|^2. \qquad (5)$$

It can be shown that the unique solution to (5) is $\hat{M}_1 = \hat{P}_1^\top A \hat{P}_1$ (see Appendix B).

**Stage 3: Restore $B_\tau$ for $\tau$-hop control.** In this step, we restore the $B_\tau$ that quantifies the "effective component" of control inputs restricted to $E_\mathrm{u}$ (see Section 3.1.2 for detailed discussion). Note that equation (2) can be rewritten in terms of $y_{1,t}$ as

$$y_{1,t_i+\tau} = M^\tau y_{1,t_i} + \Delta_\tau y_{2,t_i} + B_\tau u_{t_i}.$$

Hence, for the purpose of estimation, we simply ignore the $\Delta_\tau$ term, and take the $i^\mathrm{th}$ column as

$$\hat{b}_i \leftarrow \frac{1}{\|u_{t_i}\|} \left( \hat{P}_1^\top x_{t_i+\tau} - \hat{M}_1^\tau \hat{P}_1^\top x_{t_i} \right),$$

where $u_{t_i}$ is parallel to $e_i$, and the magnitude of $u_{t_i}$ is set to be large enough as $\alpha \|x_{t_i}\|$ to amplify its effect so that the estimation error of $A$ is comparatively negligible. Here we introduce an adjustable constant $\alpha$ to guarantee that the $E_\mathrm{u}$-component still constitutes a non-negligible proportion of the state after injecting $u_{t_i}$, so that the iterative restoration of columns could continue.

It is evident that the ignored $\Delta_\tau P_2^\top x_{t_i}$ term will introduce an extra estimation error. Since $\Delta_\tau$ contains a factor of $M_1^{\tau-1}\Delta$ that explodes with respect to $\tau$, this part can only be bounded if $\frac{\|P_2^\top x_{t_i}\|}{\|x_{t_i}\|}$ is sufficiently small. For this purpose, we introduce $\omega$ heat-up steps (running in open loop with 0 control input) to reduce the ratio to an acceptable level, during which time the projection of state onto $E_\mathrm{u}^\perp$ automatically diminishes over time since $\rho(M_2) = |\lambda_{k+1}| < 1$.

**Stage 4: Construct a $\tau$-hop stabilizing controller $K$.** Finally, we can design a controller that cancels out $M_1^\tau$ in the $\tau$-hop system. As mentioned in Section 3.1.2, we shall focus on the case where $B$ is an $n$-by-$k$ matrix for the sake of exposition (the case for general $B$ will be discussed in Appendix C). The invertibility of $B_\tau$ can be guaranteed under certain conditions (Assumption 4.3$'$); further, $\hat{B}_\tau$ is also invertible as long as it is close enough to $B_\tau$. In this case, the $\tau$-hop stabilizing controller can be simpliy designed as $\hat{K}_1 = -\hat{B}_\tau^{-1}\hat{M}_1^\tau$ in $y$-coordinates where we replace $B_\tau$ and $M_1$ in (4) with their estimates. When we return to the original $x$-coordinates, the controller becomes $\hat{K} = -\hat{B}_\tau^{-1}\hat{M}_1^\tau \hat{P}_1^\top$. Note that $\hat{K}$ (and $\hat{K}_1$) appears with a hat to emphasize the use of estimated projector $\hat{P}_1$, which introduces an extra estimation error to the final closed-loop dynamics.

It is evident that the algorithm terminates in $t_0 + k(1 + \omega + \tau)$ time steps. In the next section, we show how to choose the parameters to guarantee both stability and sub-linear sample complexity.

Finally, we remark that, although for the ease of exposition we have assumed here the instability index $k$ is known, it is fine to use an estimate of $k$ that is larger than its true value in practice — i.e., the algorithm still outputs a stabilizing controller since the performance analysis only relies on the ratio between eigenvalues and the stability of $\lambda_{k+1}$, and the complexity only suffers little if the guess of $k$ is close to its true value.

## 4 Stability Guarantee

In this section, we formally state the assumptions and show the sample complexity for the proposed algorithm to find a stabilizing controller. Our first assumption is regarding the spectral properties of $A$, where we require all eigenvalues to appear without multiplicity (so that we can learn a complete basis of each eigenspace), and marginally stable eigenvalues (i.e., those with moduli 1) are eliminated (so that eigenspaces are either stable or unstable). We would like to point out that it is common practice (e.g., [50]) to discuss marginally stable eigenvalues separately, since it obscures the distinction between stable and unstable components and is thus technically challenging.

**Assumption 4.1** (Spectral Property)**.** *$A$ is diagonalizable with instability index $k$, with distinct eigenvalues $\lambda_1, \cdots, \lambda_n$ satisfying $|\lambda_1| \geq |\lambda_2| \geq \cdots \geq |\lambda_k| > 1 > |\lambda_{k+1}| \geq \cdots \geq |\lambda_n|$.*

The assumption is mild in the sense that matrices satisfying Assumption 4.1 are dense in $\mathbb{R}^{n \times n}$, and our final complexity bound only depends logarithmically on the condition number of eigenvectors

$\kappa_{\mathrm{e}}(A)$ and the eigen-gap $\lambda_k/\lambda_{k+1}$ (see Theorem 4.1 and the discussion below). Thus any matrix $A$ that violates Assumption 4.1 can be handled via small perturbations.

Our second assumption is regarding how to choose the initial state, which again is standard. The initialization must be randomized to eliminate the coincidence where $x_0$ has zero (oblique) projection onto some eigenvector $v_i$, in which case we cannot learn about $v_i$ and thus $D$ is not invertible.

**Assumption 4.2** (Initialization). *The initial state of the system is sampled uniformly at random on the unit hyper-sphere surface in $\mathbb{R}^n$.*

Lastly, we assume the system to be $(d, \sigma)$-strongly controllable, which is standard in literature.

**Assumption 4.3** (($\nu, \sigma$)-Strong Controllability). *The system is $(\nu, \sigma)$-strongly controllable; i.e., $\sigma_{\min}(C_\nu) > \sigma$, where $C_\nu := [A^{\nu-1}B \; A^{\nu-2}B \; \cdots \; AB \; B]$ is the $\nu$-step controllability matrix.*

Above are all the assumptions we need. However, we remind the readers that, when we introduce the $\tau$-hop controller design in Section 3.1.2, $B$ is assumed to have $k$ columns and certain assumptions are needed to guarantee the invertibility of $B_1$. Indeed, for the ease of exposition, we first consider this special case in presenting our main result (Theorem 4.1) below, where we impose the following Assumption 4.3′ regarding the controllability within the unstable subspace $E_{\mathrm{u}}$ instead of the more general Assumption 4.3 (recall that $R_1$ is defined in the $E_{\mathrm{u}} \oplus E_{\mathrm{s}}$-decomposition in Section 3.1.1). Discussions on how to handle the more general Assumption 4.3 via a transformation to the special case (where Assumption 4.3′ holds) are deferred to Appendix C.

**Assumption 4.3′** ($c$-Effective Control in Unstable Subspace). $B \in \mathbb{R}^{n \times k}$, $\sigma_{\min}(R_1 B) > c\|B\|$.

Note that Assumption 4.3′ has a clear intuition — every direction in the unstable subspace receives at least a proportion of $c$ from the influence of any control input. This assumption is reasonable in that, if $\sigma_{\min}(R_1 B) \approx 0$, the control input $u$ has to be very large to push the state along the direction corresponding to the smallest singular value, which could induce excessively large control cost. We can also interpret the lower bound on $\sigma_{\min}(R_1 B)$ as a special case of Assumption 4.3 (i.e., $(1, c\|B\|)$-strong controllablility). Details can be found in Appendix C.

In the following we present the main performance guarantees for our algorithm.

**Theorem 4.1** (Main Theorem). *Given a noiseless LTI system $x_{t+1} = Ax_t + Bu_t$ subject to Assumptions 4.1, 4.2 and 4.3′, and additionally $|\lambda_1|^2|\lambda_{k+1}| < |\lambda_k|$, by running LTS$_0$ with parameters*

$$\tau = O(1), \; \omega = O(\ell \log k), \; \alpha = O(1), \; t_0 = O(k \log n)$$

*that terminates within $t_0 + k(1 + \omega + \tau) = O(k \log n)$ time steps, the closed-loop system is exponentially stable with probability $1 - O(k^{-\ell})$ over the initialization of $x_0$ for any $\ell \in \mathbb{N}$. Here the big-O notation only shows dependence on $k$ and $n$, while hiding parameters like $|\lambda_1|$, $|\lambda_k|$, $|\lambda_{k+1}|$, $\|A\|$, $\|B\|$, $c$, $\alpha$, $\xi$ (recall that $E_{\mathrm{u}}^{\perp}$ and $E_{\mathrm{s}}$ are $\xi$-close), $\chi(\hat{L}_\tau)$ (see Lemma D.1), and $\zeta_\varepsilon(\cdot)$ (see Lemma G.1), and details can be found in equations (41) through (46).*

Theorem 4.1 shows the proposed LTS$_0$ algorithm can find a stabilizing controller in $\tilde{O}(k)$ steps, which incurs a state norm of $2^{\tilde{O}(k)}$, significantly smaller than the state-of-the-art $2^{\Theta(n)}$ in the $k \ll n$ regime. We would like to point out that this does not violate the lower bound shown in [15], since the state norm degenerates to $2^{\Theta(n)}$ when $k = \Theta(n)$, and might degrade arbitrarily for systems with adversarially designed parameters. Still, for a large proportion of systems with $k \ll n$ and favorable constants, our algorithm achieves better performance than the naive ones. The theoretical result is also verified by numerical experiments, the details of which can be found in Appendix H.

**Discussion on constants.** Curious readers can refer to Appendix G (equations (41) through (46)) for detailed expressions of the constants hidden behind the big-O notation in the theorem; Table 1 also summarizes all instance-specific constants appearing in the bound. Here we provide a brief overview how the bound depends on the system parameters. It is evident that, for a system with larger $\xi$ (i.e., when $E_{\mathrm{u}}$ and $E_{\mathrm{s}}$ are "less orthogonal" to each other) or smaller $c$ (i.e., when it costs more to control the unstable subspace), we see a larger $\tau$ in (41), a smaller $\alpha$ in (43), and larger $t_0$ and $\omega$ in (45) and (46), respectively, which altogether incur a larger constant term in the sample complexity. This is in accordance with our intuition of the state space decomposition and Assumption 4.3′, respectively.

The bound also relies heavily on the spectral properties of $A$. The constraint $|\lambda_1|^2|\lambda_{k+1}| < |\lambda_k|$ ensures validity of (41), which is necessary for cancelling out the combined effect of non-orthogonal

subspaces $E_\mathrm{u}$ and $E_\mathrm{s}$ (resulting in $\Delta_\tau$ in the top-right block) and inaccurate basis $\hat{P}_1$ (resulting in projection error in the bottom-left block) — a system with larger ratio $|\lambda_1|^2|\lambda_{k+1}|/|\lambda_k|$ suffers from more severe side-effects, and thus requires a larger $\tau$ and a higher sample complexity. Nevertheless, we believe that this assumption is not essential, and we leave it as future work to relax it.

Another important parameter is the eigen-gap $|\lambda_k|/|\lambda_{k+1}|$ around 1 that determines how fast the stable and unstable components become separable in magnitude when the system runs in open loop, which is utilized in the $t_0$ initialization steps of Stage 1 and $\omega$ heat-up steps of Stage 3. Consequently, a system with smaller eigen-gap $|\lambda_k|/|\lambda_{k+1}|$ requires a larger $t_0$ (see (10)) and $\omega$ (see (46)) and therefore a higher sample complexity.

The condition number of eigenvectors $\kappa_\mathrm{e}(A)$ also contributes to the bound of $t_0$, the number of initialization steps. It is intuitive that, a large $\kappa_\mathrm{e}(A)$ indicates less orthogonal eigenspaces, which in turn requires a more distinct separation among the magnitudes of different eigen-components of $x_{t_0}$, so that the stable components interfere less with the unstable ones.

Finally, we would like to point out that all these quantities appear in the bound as *logarithmic* terms, indicating that the sample complexity only degrades mildly when the constants become worse.

**A warm-up case.** Despite the generality of Theorem 4.1, its proof involves technical difficulties. In Theorem 4.2, we include results for the special case where $A$ is real symmetric, which leads to a simpler choice of algorithm parameters and a cleaner sample complexity bound.

**Theorem 4.2.** *Given a noiseless LTI system $x_{t+1} = Ax_t + Bu_t$ subject to Assumptions 4.1, 4.2 and 4.3′ with symmetric $A$, by running LTS$_0$ with parameters $\tau = 1$, $\omega = 0$, $\alpha = 1$, $t_0 = O(k \log n)$ that terminates within $t_0 + k(1 + \omega + \tau) = O(k \log n)$ time steps, the closed-loop system is exponentially stable with probability $1$ over the initialization of $x_0$. Here the big-O notation only shows dependence on $k$ and $n$, while hiding parameters like $|\lambda_1|$, $|\lambda_k|$, $|\lambda_{k+1}|$, $\|A\|$, $\|B\|$, $c$, and $\chi(\hat{L}_1)$ (see Lemma D.1), and details can be found in equation (18).*

Although Theorem 4.2 takes a simpler form, its proof still captures the main insight of our analysis. For this reason, we use the proof of Theorem 4.2 as a warm-up example in Appendix F before we present the proof ideas of the main Theorem 4.1.

# 5 Proof Outline

In this section we will give a high-level overview of the key proof ideas for the main theorems. The full proof details can be found in Appendices E, F and G as indicated below.

**Proof Structure.** The proof is largely divided into two steps. In Step 1, we examine how accurate the learner estimates the unstable subspace $E_\mathrm{u}$ in Stage 1 and 2. We will show that $\Pi_1$, $P_1$ and $M_1$ can be estimated up to an error of $\delta$ within $t_0 = O(k \log n - \log \delta)$ steps. In Step 2, we examine the estimation error of $M_1$ and $B_\tau$ in Stage 2 and 3 (and thus $\hat{K}_1$), based on which we will eventually show that the $\tau$-hop controller output by Algorithm 1 makes the system asymptotically stable. The proof is based on a detailed spectral analysis of the closed-loop dynamical matrix.

**Overview of Step 1.** To upper bound the estimation errors in Stage 1 and 2, we only have to notice that the estimation error of $\Pi_1$ completely captures how well the unstable subspace is estimated, and all other bounds should follow directly from it. The bound on $\|\Pi_1 - \hat{\Pi}_1\|$ is shown in Theorem 5.1, together with a bound on $\|P_1 - \hat{P}_1\|$ presented in Corollary 5.2.

**Theorem 5.1.** *For a noiseless linear dynamical system $x_{t+1} = Ax_t$, let $E_\mathrm{u}$ be the unstable subspace of $A$, $k = \dim E_\mathrm{u}$ be the instability index of the system, and $\Pi_1$ be the orthogonal projector onto subspace $E_\mathrm{u}$. Then for any $\varepsilon > 0$, by running Stage 1 of Algorithm 1 with an arbitrary initial state that terminates in $(t_0 + k)$ time steps, where*

$$t_0 = O\left(\frac{k \log n - \log \varepsilon + \log \kappa_\mathrm{e}(A)}{2 \log \frac{|\lambda_k|}{|\lambda_{k+1}|}}\right),$$

*the matrix $D^\top D$ is invertible with probability $1$ (where $D = [x_{t_0+1} \cdots x_{t_0+k}]$) and in such cases we shall obtain an estimated $\hat{\Pi}_1 = D(D^\top D)^{-1}D^\top$ with error $\|\hat{\Pi}_1 - \Pi_1\| < \varepsilon$.*

**Corollary 5.2.** *Under the premises of Theorem 5.1, for any orthonormal basis $\hat{P}_1$ of $\mathrm{col}(\hat{\Pi}_1)$ (where $\hat{\Pi}_1$ is obtained by Algorithm 1), there exists a corresponding orthonormal basis $P_1$ of $\mathrm{col}(\Pi_1)$, such that $\|\hat{P}_1 - P_1\| < \sqrt{2k}\varepsilon =: \delta$, $\|\hat{M}_1 - M_1\| < 2\|A\|\delta$.*

The proofs are deferred to Appendix E due to limited length.

**Overview of Step 2.** To analyze the stability of the closed-loop system, we shall first write out the closed-loop dynamics under the $\tau$-hop controller. Recall in Section 3.1.2 we have defined $\tilde{u}_s, \tilde{x}_s, \tilde{y}_s$ to be the control input, state in $x$-coordinates, and state in $y$-coordinates in the $\tau$-hop control system, respectively. Using these notations, the learned controller can be written as

$$\tilde{u}_s = \hat{K}\tilde{x}_s = \hat{K}_1 \hat{P}_1^\top P \tilde{y}_s = \begin{bmatrix} \hat{K}_1 \hat{P}_1^\top P_1 \\ \hat{K}_1 \hat{P}_1^\top P_2 \end{bmatrix} \tilde{y}_s$$

in $y$-coordinates (as opposed to $\hat{K}_1 \tilde{y}_s$). Therefore, the closed-loop $\tau$-hop dynamics should be

$$\tilde{y}_{s+1} = \begin{bmatrix} M_1^\tau + P_1^\top A^{\tau-1} B \hat{K}_1 \hat{P}_1^\top P_1 & \Delta_\tau + P_1^\top A^{\tau-1} B \hat{K}_1 \hat{P}_1^\top P_2 \\ P_2^\top A^{\tau-1} B \hat{K}_1 \hat{P}_1^\top P_1 & M_2^\tau + P_2^\top A^{\tau-1} B \hat{K}_1 \hat{P}_1^\top P_2 \end{bmatrix} \begin{bmatrix} \tilde{y}_{1,s} \\ \tilde{y}_{2,s} \end{bmatrix} =: \hat{L}_\tau \tilde{y}_s, \quad (6)$$

and we will show it to be asymptotically stable (i.e., $\rho(\hat{L}_\tau) < 1$). Note that $\hat{L}_\tau$ is given by a 2-by-2 block form, we can utilize the following lemma to assist the spectral analysis of block matrices, the proof of which is deferred to Appendix D.

**Lemma 5.3** (Block Perturbation Bound). *For 2-by-2 block matrices $A = \begin{bmatrix} A_1 & \mathbf{0} \\ \mathbf{0} & A_2 \end{bmatrix}$, $E = \begin{bmatrix} \mathbf{0} & E_{12} \\ E_{21} & \mathbf{0} \end{bmatrix}$, the spectral radii of $A$ and $A + E$ differ by at most $|\rho(A + E) - \rho(A)| \leq \chi(A + E)\|E_{12}\|\|E_{21}\|$, where $\chi(A + E)$ is a constant (see Appendix D).*

The above lemma shows a clear roadmap for proving $\rho(\hat{L}_\tau) < 1$. First, we need to guarantee stability of the diagonal blocks — the top-left block is stable because $\hat{K}_1$ is designed to (approximately) eliminate it to zero (which requires the estimation error bound on $B_\tau$), and the bottom-right block is stable because it is almost $M_2^\tau$ with a negligible error induced by inaccurate projection. Then, we need to upper-bound the norms of off-diagonal blocks via careful estimation of factors appearing in these blocks. Complete proofs for both cases can be found in Appendices F and G, respectively.

## 6 Conclusions

This paper provides a new perspective into the learn-to-stabilize problem. We design a novel algorithm that exploits instance-specific properties to learn to stabilize an unknown LTI system on a single trajectory. We show that, under certain assumptions, the sample complexity of the algorithm is upper bounded by $O(k \log n)$, which avoids the $2^{\Theta(n)}$ state norm blow-up of existing methods in the $k \ll n$ regime. This work initiates a new direction in the learn-to-stabilize literature, and many interesting and challenging questions remain open, including handling noises, eliminating the assumptions on spectral properties, and developing better ways to learn the unstable subspace.

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
