# OpenReview forum: "On the Sample Complexity of Stabilizing LTI Systems on a Single Trajectory"
_NeurIPS.cc/2022/Conference — NeurIPS 2022 Accept_

### Official Review · Reviewer_fVVW · 2022-07-11

**Rating:** 5
**Confidence:** 4
**Soundness:** 3 good
**Presentation:** 3 good
**Contribution:** 2 fair

**Summary:**

The paper studies the number of samples for learning to stabilize a noiseless linear control system. Under a set of assumptions, it is shown that with O(k log n) samples the system can be stabilized, where n is the state dimension and k is the instability dimension. An algorithm is presented for that purpose too.

**Questions:**

See above

**Limitations:**

See above.

**Strengths And Weaknesses:**

Strengths:
The paper discusses different steps and explain them. Also, the technicalities and the assumptions are stated in a way the reader can follow.

Weaknesses:
I like to see how the authors can discuss the following items.
- Overall, the writing is repetitive in multiple places.
- I think connections to some relevant stabilization algorithms in the literature (mostly subject to stochasticity and/or using that) are missing.
- In the abstract, 'instability index' needs a (high level) definition, and the rates are not correctly written (the current one seems like k log k).
- The assumptions are strong. The open loop matrix is diagonalizable, does not have any unit eigenvalue and even has a significant eigen-gap (Theorem 4.1), the unstable and stable eigenspaces are assumed almost orthogonal, the dimension of the control input is exactly the same as the number of unstable eigenvalues, etc.
- On top of the above, having an undisturbed dynamics is a very important and strong assumption. Although, the authors correctly claim that it is not consequential in terms of stability, it is important for learning the unknown matrices by letting the observations be noisy or noiseless, and also as mentioned in line 196, the stable modes all die out in undisturbed systems. Importantly, most of the assumptions here are not needed in the existing literature, and some are proven to be satisfiable through designing the control input properly, rather than being given.
- Along the same axis, for extending to stochastic systems and having high probability results, it is not clear how this work helps as the authors claim in line 120.
- Some sample presentation points to improve are as follows. In Def 3.1, I think orthonormality of the matrices is crucial and needs to be emphasized and mentioned explicitly. In line 5 of Algorithm 1, I think it should be 'ortho-normalized'. The range of i is not clear in line 11. Line 258 and invertibility needs more concrete explanation. R_1 in Assumption 4.3 is better to be referred to where defined. I could not find the 'recall' in Theorem 4.1, and it might be added to the assumptions.

---

> ### Author Response · Authors · 2022-08-02
> **Response to Reviewer fVVW**
>
> Thank you for your insightful comments and suggestions!
>
> Our paper provides a new perspective on the learn-to-stabilize problem by providing instance-specific bounds that break the worst-case $2^{\Omega(n)}$  lower bound suggested by the literature. Our results rely on decomposing the state space into stable and unstable components and learning only the unstable component, which, to the best of knowledge, is a novel approach. We appreciate your comments on the limitation of the assumptions. As will be addressed point-by-point below, some of these assumptions can be easily relaxed based on the current techniques, while other assumptions require future work. We impose these assumptions because breaking the worst case $2^{\Omega(n)}$ lower bound is already a challenging problem, and hopefully our results can serve as an initial step and inspire future work that can push the boundary on the learn-to-stabilize problem.
>
> Please find below our point-by-point responses.
>
>
> **(Presentation suggestions)**
> Thanks for your insightful suggestions on presentation! We will try to modify them accordingly in the revision (e.g., trimming some repeated paragraphs). We will also briefly define $k$ in the abstract.
>
>
> **(Connections with existing algorithms)**
> We already have some brief discussions on existing algorithms in the introduction (lines 32 through 52). Further, reviewer JYWD provides some good references like [Talebi2020], [Tsiamis2022], etc. [Talebi2020] also simplifies system identification via projection onto some lower-dimensional subspace, but it considers a different scenario that only stabilizes the system with a fixed initial state $x_0$. [Tsiamis2022] studies the complexity of stabilization with a focus on under-actuated systems. We will add more discussion in the revision.
>
>
> **(Notation $\tilde{O}(k)$)**
> Thanks for pointing this out and we agree that the $\tilde{O}(k)$ notation may be confusing here, since it implicitly assumes $\log n$ and $\log k$ to be in the same order. We will be more precise and write $O(k \log n)$ instead in the revision.
>
>
> **(Handle stochastic noise)**
> We focus on the noiseless case because we would like to identify the minimum information needed to learn to stabilize a dynamical system, a task that is already challenging in the noiseless setting. We agree that the noisy setting is important, and we think there is a path to generalize our results to it.
>
> Algorithm-wise, the potential modifications lie in the identification part of the algorithm, namely Stage 1 \& 2. For Stage 1, to eliminate the stochastic influence of noises, $D$ should sample data from a window longer than $k$ steps, and then we calculate the top $k$ eigenvectors of $D D^\top$ to estimate the unstable subspace $\mathrm{col}(\hat{P}_1)$. Similarly, for Stage 2, a longer sampling window is needed for the least squares.
>
> Analysis-wise, we expect that techniques from system identification literature (see, e.g. [Rakhlin2018]) can be applied here. Specifically, our estimation problem is similar to the “explosive” case in [Rakhlin2018] where it is assumed all the eigenvalues of the matrix $A$ is unstable. In our case, we estimate a component $M_1$ of $A$ whose eigenvalues are also unstable. [Rakhlin 2018] shows that for the explosive case, the estimation error actually decays exponentially with the number of samples, which, when applied to our setting, means the number of extra steps needed for Stage 1 \& 2 can be only logarithmic. Having said that, we also acknowledge that applying the techniques in [Rakhlin2018] requires detailed and lengthy calculations. We believe it is best left as future work and we will add these discussions in the revision.

---

> > ### Author Response · Authors · 2022-08-02
> > **Response to Reviewer fVVW (cont'd) and References**
> >
> > **(On the assumption $|\lambda_1|^2 |\lambda_{k+1}| < |\lambda_k|$)**
> > While this assumption seems strong, we point out that it is necessary for establishing theoretical bounds for our algorithm, and that in some special cases it could potentially be removed.
> >
> > The core reason why the assumption is needed is that the stable and unstable components are coupled with each other due to the non-orthogonality of $E_{\mathrm{u}}$ and $E_{\mathrm{s}}$, which makes the analysis of closed-loop stability challenging. For example, as can be shown in eq. (1) (on page 4, below line 166), any attempt by the controller to reduce the unstable component will inevitably increase the stable component, which then further propagates to the unstable component due to the off-diagonal block $\varDelta$ in eq. (1). To counter such side effects, we have to carefully design the $\tau$-hop controller to inhibit the coupling, and in doing so we do require certain assumptions on the eigen-gap.
> >
> > Relaxing this assumption is important and is left as future work. For example, based on our current techniques, this assumption can be relaxed when $\xi$ is sufficiently small (i.e., when $E_{\mathrm{u}}$ and $E_{\mathrm{s}}$ are almost orthogonal). In this setting, the off-diagonal block $\varDelta$ in (1) is close to $0$ and we don't need to use the $\tau$-hop controller. Hence we expect a cleaner bound (similar to that in the warm-up case) that does not depend on this eigen-gap assumption. Another potential way to relax the assumption is to avoid fully cancelling out the unstable component. Rather, we only have to find a stabilizing controller to *stabilize* the unstable component, which leaves some space to relax the assumption. All these are interesting directions and we leave them as future work.
> >
> >
> > **(Other discussions on assumptions)**
> > + *(On $\xi$-closeness property of $E_{\mathrm{u}}$ and $E_{\mathrm{s}}$)*
> > We would like to clarify that we do *not* assume that the subspaces $E_{\mathrm{u}}$ and $E_{\mathrm{s}}$ are almost orthogonal — they are assumed to be $\xi$-close, but $\xi$ can take any value in $(0,1)$, and a larger $\xi$ (meaning $E_{\mathrm{u}}$ and $E_{\mathrm{s}}$ are less orthogonal) will only worsen the upper bound in a logarithmic way (see eq. (41) through (46)). See the discussion between lines 311 and 315, and also Table 1 at the end of Appendix G.
> >
> > + *(Marginally stable eigenvalues)*
> > Marginally stable eigenvalues are known to be a tricky issue, and it is common practice (see, e.g., [Rakhlin2018]) to discuss it separately. We will acknowledge this point in the revision.
> >
> > + *(On Assumption 4.3 and the transformation)*
> > We only assume $k=m$ for ease of presentation, and the reduction from cases where $k \neq m$ can be found in Appendix C. Specifically, we show a reduction from the general case to the $k=m$ case by concatenating states in $d$ consecutive steps as a new single state, where $d \leq k$. This reduction will only worsen the norms of $A$ and $B$ by a factor of $\Vert A \Vert^d O(d)$, and the sample complexity bound will contain an additional $O(k \log k)$ term since $\Vert A \Vert$ and $\Vert B \Vert$ only appear in logarithms (see eq. (41) through (46)).
> >
> >
> > **References**
> >
> > [Rakhlin2018] Tuhin Sarkar and Alexander Rakhlin. Near optimal finite time identification of arbitrary linear dynamical systems, 2018, *arXiv: arXiv preprint 1812.01251*.
> >
> > [Talebi2020] S. Talebi, S. Alemzadeh, N. Rahimi and M. Mesbahi. Online Regulation of Unstable Linear Systems from a Single Trajectory, *2020 59th IEEE Conference on Decision and Control (CDC)*, 2020, pp. 4784-4789.
> >
> > [Tsiamis2022] Anastasios Tsiamis, Ingvar M Ziemann, Manfred Morari, Nikolai Matni, George J. Pappas. Learning to Control Linear Systems can be Hard. *Proceedings of Thirty Fifth Conference on Learning Theory (PMLR)*, 178:3820-3857, 2022.

---

> > > ### Comment · Reviewer_fVVW · 2022-08-08
> > > **Re**
> > >
> > > This reviewer appreciate the comments of the authors. However, the responses do not seem convincing. The authors more or less agree that addressing the restrictions of the technical assumptions is not straightforward. Also, the discussion on stochastic systems is not convincing as generalizing the proposed analysis is clearly non-trivial. The reference to explosive systems is also inaccurate, as in the case of having both explosive and non-explosive modes, their interaction is a fundamental obstacle in the analysis. Furthermore, unit eigenvalues are shown to be to some extend avoidable, e.g. in [44], and causing no problem for the analysis. The reviewer agree that some of the generalizations can be left as a future work, but it does not refute that the current work is incomplete. Accordingly, the reviewer prefers to see if the authors can discuss further so that the rating need to be updated.

---

> > > > ### Author Response · Authors · 2022-08-09
> > > > **Response to follow-up comment**
> > > >
> > > > Thanks for your additional comments!
> > > >
> > > > 1. (technical assumptions) We would like to point out that the implicit assumption $k = m$ in Assumption 4.3 is actually removed using the method in Appendix C. Also, Assumption 4.3 in Appendix C can be further relaxed to the controllability of the original system (see the response to reviewer JYWD, cont’d part 1, for a detailed explanation). And further, we think the first two assumptions are very standard in literature. For a kind reminder, as we have discussed in the initial response, we do not assume almost-orthogonality between the stable/unstable subspaces. Given the above comments, at this point, we believe that these assumptions are standard and mild.
> > > >
> > > > 2. (noisy case) Regarding our reference to the explosive case, what we meant is that in Stage 2 of the algorithm, since we restrict our scope to the unstable subspace and only learn the $M_1$ matrix, the projected trajectory used in the least squares would behave as if they were generated by $M_1$ (which only has unstable eigenvalues). This is similar to the explosive case in [Rakhlin2018].
> > > >
> > > > 3. (unit eigenvalue issue) We agree that [44] applies a technique that uses a sequence of random excitation matrices $L_1, \cdots, L_k$, so that marginal systems (i.e., with eigenvalue on the unit circle) can be converted into non-marginal systems with high probability. Their technique works well for their algorithm directly based on least squares and yields a $O(n \log n)$ sample complexity. However, we use a different system identification approach that tries to avoid identifying the whole system. For this purpose, we need full control over the selection of excitation matrices, and therefore the least-square-based algorithm and analysis do not apply here. We also point out that, the system identification paper that [44] relies on also circumvents the marginally stable case.
> > > >
> > > > 4. (contribution of this paper) Lastly, we emphasize that the overall framework proposed in this framework is novel and presents an interesting new perspective on the learning-to-stabilize problem. We believe this is already a significant contribution, and it opens the door for many future approaches that can be based on the same stable/unstable decomposition. While there are certain technical assumptions, as the reviewer agrees, some of the assumptions are challenging to generalize and can thus be left as future work. We will incorporate all suggestions from the reviewers and add more discussions in revision regarding the assumptions.

---

### Official Review · Reviewer_JYWD · 2022-07-11

**Rating:** 6
**Confidence:** 5
**Soundness:** 3 good
**Presentation:** 2 fair
**Contribution:** 2 fair

**Summary:**

The paper studies the problem of stabilization from data in the case of linear control systems, in the noiseless setting. A new algorithm is proposed, where instead of identifying the whole state space, it is sufficient to identify only the unstable part. Under certain assumptions, this is sufficient for designing a stabilizing control, requiring fewer samples than the case when everything is identified. The paper provides some sample complexity bounds with high probability, where the randomness comes from the initial condition. If some special conditions are satisfied, then the complexity is linear with k, i.e. the total number of unstable eigenvalues, and depends logarithmically on n, i.e. the dimension of the state space.

​

**Questions:**

Please compare with the two papers listed above, Talebi, Alemzadeh, Rahimi, Mesbahi, "Online Regulation of Unstable Linear Systems from a Single Trajectory", CDC 2020 and Tsiamis, Ziemann, Morari, Matni, Pappas "Learning to control linear systems can be hard", 2022.
​
There should be more intuition/discussion about what types of systems satisfy Assumption 4.3 , k=m.
​
Please provide more details for the case of general matrices B (when m is less than k) and clarify Section C.  What would Assumption 4.3 look like in this setting? How would the result of Theorem 4.1 depend on the index d? To my understanding, in the worst case, we might need to unroll the state d=theta(n), not theta(k/m), times to achieve controllability and stabilize the system. I believe, this d can be as high as the controllability index of the system, which in the worst case can be linear with n, even if the number of unstable modes is as small as one (unless further assumptions are imposed).   For example, see system (14) in Tsiamis, Ziemann, Morari, Matni, Pappas "Learning to control linear systems can be hard", 2022. For that system, tilde B (projected on the unstable subspace) would be equal to zero unless d=n-1.
​
There seems to be a problem with the invertibility of matrix D'D. In particular, if matrix A is singular, then D'D should also be singular. Probably the results still go through since a projection is considered. But the proofs of section E should be updated.
​
Could you please provide more intuition why u has this particular form in line  11 of the algorithm? Why does the input scale with the norm of x? Since x is unstable, this might lead to very large control inputs. Please clarify that in the text since this seems to be another limitation of the algorithm. The naive algorithm would work with small inputs.
​
Please provide more discussion on which assumptions seem necessary and which assumptions seem removable.

**Limitations:**

See comment above.


**Strengths And Weaknesses:**

As mentioned in the summary, if certain special conditions hold, then the complexity is linear with k, i.e. the total number of unstable eigenvalues, and depends logarithmically on n, i.e. the dimension of the state space.
This is a novel and interesting result that was not established before. Hence, the benefit of this algorithm is that we can stabilize the system much faster compared to the standard algorithm, where we identify the whole state space.
It would be interesting to compare the results of this paper to the paper by Talebi, Alemzadeh, Rahimi, Mesbahi, "Online Regulation of Unstable Linear Systems from a Single Trajectory", CDC 2020, where a similar topic is explored, under similar assumptions.
​
The main weakness  of the paper is that it relies on several restrictive assumptions that should be explained upfront and in more detail. More intuition should be provided; for example it would help to provide a discussion on what types of systems satisfy these assumptions. Also it should be explained whether the algorithm still works or fails in the absence of these assumptions.
The most crucial assumption is that all unstable modes are directly driven/controllable by the input matrix B (Assumption 4.3 along with k=input dimension m). In other words, it is assumed that we have a lot of inputs and all of them directly affect any possible unstable mode. This assumption is quite restrictive since it rules out the possibility of underactuated systems, systems with network structure or systems with integrator-like structure, which in general are hard to control, see the paper by Tsiamis, Ziemann, Morari, Matni, Pappas "Learning to control linear systems can be hard", 2022.
The material of Section C does not explicitly address the case when we might have k larger than the number of inputs (m less than k)-see also questions.
There are three more assumptions that are somewhat restrictive yet crucial for obtaining the result. i) There should be no noise in the system, ii) The initial condition should be non-zero, iii) the stable eigenvalues cannot be arbitrary; they should be small enough to cancel the effect of the unstable ones (in the cross-term entries of matrix M tau). The paper provides technical reasons why we need them, but it would be nice to also see some discussion whether they are necessary or not. For example, having stochastic noise might affect sample complexity in a dramatic way; in deterministic systems it is always possible to identify everything with zero error and finite samples while this is not possible in the case of stochastic systems. The assumption of the stable part cancelling the unstable one seems artificial, would the algorithm work if it was removed?

​

---

> ### Author Response · Authors · 2022-08-02
> **Response to Reviewer JYWD**
>
> Thank you for your insightful comments and suggestions!
>
> Our paper provides a new perspective on the learn-to-stabilize problem by providing instance-specific bounds that break the worst-case $2^{\Omega(n)}$  lower bound suggested by the literature. Our results rely on decomposing the state space into stable and unstable components and learning only the unstable component, which, to the best of knowledge, is a novel approach. We appreciate your comments on the limitation of the assumptions. As will be addressed point-by-point below, some of these assumptions can be easily relaxed based on the current techniques, while other assumptions require future work. We impose these assumptions because breaking the worst case $2^{\Omega(n)}$ lower bound is already a challenging problem, and hopefully our results can serve as an initial step and inspire future work that can push the boundary on the learn-to-stabilize problem.
>
> Please find below our point-by-point responses.
>
>
> **(Compare against [Talebi2020])**
> We agree that [Talebi2020] also simplifies system identification via projection onto some lower-dimensional subspace. However, as we understand it, [Talebi2020] considers a different scenario that only stabilizes the system with a fixed initial state $x_0$. In fact, it only shows that if $x_0$ lies in a $k$-dim subspace, their algorithm can eliminate the unstable component *along that trajectory* in $k$ steps; however, in most cases we should not expect $x_0$ to have such property. Our approach is similar in ideas, but starting from a random $x_0$, we want to find a stabilizing controller for the system *along an arbitrary trajectory*, which is more general.
>
> Starting from a random $x_0$ makes the analysis a lot more challenging because the stable and unstable components are coupled with each other due to the non-orthogonality of $E_{\mathrm{u}}$ and $E_{\mathrm{s}}$. As can be shown in eq. (1) (on page 4, below line 166), any attempt by the controller to reduce the unstable component will inevitably increase the stable component, which then further propagates to the unstable component due to the off-diagonal block $\varDelta$ in eq. (1). To counter such side effects, we have to carefully design the controller to inhibit the coupling, which leads to the idea of $\tau$-hop controller (see Section 3.1.2).
>
> We also point out that the $k$ in [Talebi2020] is determined by the initial state, while the $k$ in our paper represents the instability index of the system, which is trajectory-independent.
>
>
> **(Compare against [Tsiamis2022])**
> We agree that there are already some worst-case lower bounds on the sample complexity of stabilization, including [Tsiamis2022], [Chen2021], etc. However, we point out that these bounds only show that all *general-purpose* controllers are doomed to suffer a *worst-case* regret of $2^{\Omega(n)}$, but they do not rule out the possibility that we can achieve better regret *in specific systems* (e.g., where $k \ll n$). In fact, the lower bounds in these papers are the very reason that motivates us to seek instance-specific bound that can avoid the exponential blow up in $n$.
>
> Another difference is that [Tsiamis2022] only considers marginally stable systems, while we consider unstable systems.
>
> We also mention that [Tsiamis2022] was first publicly available after the NeurIPS deadline.
>
> Having said all these, we think the results in [Tsiamis2022] are very important and relevant to our paper. We will add more discussions on the connections in the revision.

---

> > ### Author Response · Authors · 2022-08-02
> > **Response to Reviewer JYWD (cont'd, part 1)**
> >
> > **(On Assumption 4.3 and the transformation)**
> > Here we can actually show $d \leq k$ when the original system is controllable. Suppose after the transformation (see Appendix C), the “transformed $R$ matrix” (in the $E_{\mathrm{u}} \oplus E_{\mathrm{s}}$ decomposition, see line 582) becomes $\tilde{R} = \mathrm{diag}(*, \cdots, *, R)$, and $\tilde{R}_1 = [0 ~ \cdots ~ 0 ~ R_1]$. Also recall that $R_1 A = N_1 R_1$ (i.e., $R_1$ consists of unstable left-eigenvectors of $A$). Therefore, to check whether the transformed system satisfies Assumption 4.3, we directly calculate
> > $$\tilde{R}_1 \tilde{B} = [R_1 A^{d-1} B ~ \cdots ~ R_1 AB ~ R_1 B] = [N_1^{d-1} R_1 B ~ \cdots ~ N_1 R_1 B ~ R_1 B],$$
> > which is the controllability matrix of the $k$-dimensional LTI system $(N_1, R_1 B)$. If the strong-controllability of $(N_1, R_1 B)$ holds, we will only need $d \leq k$ for $\tilde{R}_1 \tilde{B}$ to meet Assumption 4.3.
> >
> > The strong-controllability of $(N_1, R_1 B)$, on the other hand, is a direct consequence of the strong-controllability of $(A, B)$. To see this, note $(N, RB) = (Q^{-1} A Q, Q^{-1} B)$ is a similar transform of $(A, B)$ and is thus controllable. Given the block diagonal form of $(N, RB)$ (see the equation below line 586 in Appendix A.1), this also implies $(N_1,R_1B)$ is controllable. We will revise Appendix C in the revision.
> >
> > Specifically, consider the example (eq. (14) in [Tsiamis2022]) mentioned by the reviewer, we can show that in the example setting $d=1$ will satisfy the requirement (i.e., the transformation is not needed). This is because in that example, $R_1$ is a dense row vector (the left eigenvector of $A$ corresponding to eigenvalue $1$). As such, $R_1 B $ is non-zero and Assumption 4.3 is met. We think the underlying reason why the counter-example in [Tsiamis2022] does not apply to our case is that we only care about the controllability of the $k$-dimensional “unstable partial system” $(N_1, R_1 B)$, as opposed to the full system.
> >
> >
> > **(Large control inputs proportionate to $\Vert x_t \Vert$)**
> > We agree that overly large control inputs might not be practical in some systems, but they are necessary if we want to reduce the blow-up of the states. The reason is that, in the learning stage, little is known about the system and we don't have a stabilizing controller, so the state norm has to blow up at the beginning. Then, when the algorithm is learning $B$, we need to use large inputs because we want to amplify the effect of control input to make the estimation error of $A$ negligible (see eq. (44), where the upper bound of $\delta$ depends on $\alpha$).
> >
> > To some extent, this reflects the tradeoff between the blow-up of states and the blow-up of control inputs, which is largely unavoidable and is not specific to our algorithm. Our approach reduces the blow-up of states at the cost of large inputs, while the naive approach induces larger blow-up of states using smaller control inputs. This is an interesting point and we will add it to the discussion in the revision.
> >
> >
> > **(Handle stochastic noise)**
> > We focus on the noiseless case because we would like to identify the minimum information needed to learn to stabilize a dynamical system, a task that is already challenging in the noiseless setting. We agree that the noisy setting is important, and we think there is a path to generalize our results to it.
> >
> > Algorithm-wise, the potential modifications lie in the identification part of the algorithm, namely Stage 1 \& 2. For Stage 1, to eliminate the stochastic influence of noises, $D$ should sample data from a window longer than $k$ steps, and then we calculate the top $k$ eigenvectors of $D D^\top$ to estimate the unstable subspace $\mathrm{col}(\hat{P}_1)$. Similarly, for Stage 2, a longer sampling window is needed for the least squares.
> >
> > Analysis-wise, we expect that techniques from system identification literature (see, e.g. [Rakhlin2018]) can be applied here. Specifically, our estimation problem is similar to the “explosive” case in [Rakhlin2018] where it is assumed all the eigenvalues of the matrix $A$ is unstable. In our case, we estimate a component $M_1$ of $A$ whose eigenvalues are also unstable. [Rakhlin 2018] shows that for the explosive case, the estimation error actually decays exponentially with the number of samples, which, when applied to our setting, means the number of extra steps needed for Stage 1 \& 2 can be only logarithmic. Having said that, we also acknowledge that applying the techniques in [Rakhlin2018] requires detailed and lengthy calculations. We believe it is best left as future work and we will add these discussions in the revision.

---

> > > ### Comment · Reviewer_JYWD · 2022-08-03
> > > **Updated score**
> > >
> > > Thanks for the comments to the review.  I have modified the score to reflect the discussion and my improved understanding.   It is all fine to add assumptions and structure to improve/avoid general lower bounds but the proposed structure must be interesting application-wise.  At this point the proposed structure is more technically motivated so the hope is that this can provide insight for more impactful system classes in the future.

---

> > ### Author Response · Authors · 2022-08-02
> > **Response to Reviewer JYWD (cont'd, part 2) and References**
> >
> > **(Other discussion on assumptions)**
> > + It does not harm generality to set $x_0 \neq 0$, since when $x_0 = 0$ we can trivially output $u_t \equiv 0$ to guarantee stability.
> > + The algorithm also works for singular $A$s, and $D^{\top} D$ will still be invertible since it is a $k$-by-$k$ matrix (conceptually, it only depends on the full-rankness of the unstable subspace).
> >
> >
> > **References**
> >
> > [Rakhlin2018] Tuhin Sarkar and Alexander Rakhlin. Near optimal finite time identification of arbitrary linear dynamical systems, 2018, *arXiv: arXiv preprint 1812.01251*.
> >
> > [Simchowitz2018] Max Simchowitz, Horia Mania, Stephen Tu, Michael I. Jordan, and Benjamin Recht. Learning Without Mixing: Towards A Sharp Analysis of Linear System Identification, *Proceedings of the 31st Conference On Learning Theory (PMLR)*, 75:439-473, 2018.
> >
> > [Talebi2020] S. Talebi, S. Alemzadeh, N. Rahimi and M. Mesbahi. Online Regulation of Unstable Linear Systems from a Single Trajectory, *2020 59th IEEE Conference on Decision and Control (CDC)*, 2020, pp. 4784-4789.
> >
> > [Tsiamis2022] Anastasios Tsiamis, Ingvar M Ziemann, Manfred Morari, Nikolai Matni, George J. Pappas. Learning to Control Linear Systems can be Hard. *Proceedings of Thirty Fifth Conference on Learning Theory (PMLR)*, 178:3820-3857, 2022.

---

### Official Review · Reviewer_ytQe · 2022-07-15

**Rating:** 6
**Confidence:** 4
**Soundness:** 4 excellent
**Presentation:** 3 good
**Contribution:** 3 good

**Summary:**

The paper provides an algorithm for stabilizing a noiseless LTI system on a single trajectory with sample complexity scaling in the number of unstable modes of the dynamics matrix rather than the state-dimension.

**Questions:**

Noise definitely makes the analysis more challenging/messy, but is there a path to prove similar results for the noise setting?

Can the results easily be generalized to unknown $k$?

**Limitations:**

Limitation of known $k$ could be addressed directly.

**Strengths And Weaknesses:**

Strengths:
- At it's core, this is an important problem.  LTI systems are one of the most important primitive in control and exponential state blow-up in the state dimension is a critical problem in all existing work in this area as far as I know.
- Writing is generally quite clear and well organized. The assumptions are presented clearly in section 4 and for the most part are reasonable (see weaknesses).  Breakdown of the algorithm and exposition into stages also is very nice for clarity.
- Work seems technically correct and strong.

Weaknesses:
- While the algorithm is empirically justified in the presence of noise in App. H, no provable results for the noised case significantly weaken the result.  As mentioned in the paper, many existing methods first need a stabilizing controller.  This will need to be found in the presence of noise, which for some applications even needs to be adversarial.
- It appears that the instability index $k$ needs to be known for the algorithm, which probably is not reasonable in general.
- The algorithm outputs a $\tau$-hop stabilizing controller, which may not be suitable for certain downstream control tasks.
- Assumption 4.1 seems reasonable but may actually be difficult to deal with without handling noise.  Many physical systems do have repeated eigenvalues and small perturbations to the system parameters would require some form of noise tolerance.  Ideally, robustness in the $H_{\infty}$ sense would be involved as these perturbations depend on the state.


While there are limitations, I think this is result still would be a worthwhile addition to the conference.

---

> ### Author Response · Authors · 2022-08-02
> **Response to Reviewer ytQe**
>
> Thank you for your insightful comments and suggestions!
>
> Our paper provides a new perspective on the learn-to-stabilize problem by providing instance-specific bounds that break the worst-case $2^{\Omega(n)}$  lower bound suggested by the literature. Our results rely on decomposing the state space into stable and unstable components and learning only the unstable component, which, to the best of knowledge, is a novel approach. We appreciate your comments on the limitation of the assumptions. As will be addressed point-by-point below, some of these assumptions can be easily relaxed based on the current techniques, while other assumptions require future work. We impose these assumptions because breaking the worst case $2^{\Omega(n)}$ lower bound is already a challenging problem, and hopefully our results can serve as an initial step and inspire future work that can push the boundary on the learn-to-stabilize problem.
>
> Please find below our point-by-point responses.
>
>
> **(Handle stochastic noise)**
> We focus on the noiseless case because we would like to identify the minimum information needed to learn to stabilize a dynamical system, a task that is already challenging in the noiseless setting. We agree that the noisy setting is important, and we think there is a path to generalize our results to it.
>
> Algorithm-wise, the potential modifications lie in the identification part of the algorithm, namely Stage 1 \& 2. For Stage 1, to eliminate the stochastic influence of noises, $D$ should sample data from a window longer than $k$ steps, and then we calculate the top $k$ eigenvectors of $D D^\top$ to estimate the unstable subspace $\mathrm{col}(\hat{P}_1)$. Similarly, for Stage 2, a longer sampling window is needed for the least squares.
>
> Analysis-wise, we expect that techniques from system identification literature (see, e.g. [Rakhlin2018]) can be applied here. Specifically, our estimation problem is similar to the “explosive” case in [Rakhlin2018] where it is assumed all the eigenvalues of the matrix $A$ is unstable. In our case, we estimate a component $M_1$ of $A$ whose eigenvalues are also unstable. [Rakhlin 2018] shows that for the explosive case, the estimation error actually decays exponentially with the number of samples, which, when applied to our setting, means the number of extra steps needed for Stage 1 \& 2 can be only logarithmic. Having said that, we also acknowledge that applying the techniques in [Rakhlin2018] requires detailed and lengthy calculations. We believe it is best left as future work and we will add these discussions in the revision.
>
>
> **(On the assumption that $k$ should be known beforehand)**
> This is not a limiting assumption because one can either (a) use a larger $k$ (as suggested by the reviewer) or (b) learn the actual instability index $k$. We elaborate on the two approaches below.
>
> (a) It is fine to use in the algorithm a $k$ larger than the true instability index — the algorithm still outputs a stabilizing controller because our analysis only relies on the ratio between eigenvalues rather than their values, and the complexity only suffers a little if $k$ is not set to be significantly larger.
>
> (b) It is possible to determine the true instability index by the singular value decomposition of $D$ (see line 233) as follows: after the initial $t_0$ steps, we form a matrix $D^{(i)} = [x_{t_0+1} ~ \cdots ~ x_{t_0+i}]$ when $x_{t_0+i}$ is obtained; then, we calculate the singular values of the $D^{(i)}$ matrix.  Whenever $D^{(i)}$ starts to contain some singular value that is almost $0$, we know $i-1$ is exactly the instability index. The reason why this is correct is that, when $i\leq k$, all singular values of $D^{(i)}$ should be larger than $1$; when $i>k$, since the system only has $k$ unstable modes, only $k$ singular values of $D^{(i)}$ will be larger than $1$, and the rest should be close to $0$.
>
>
> **(Issues with $\tau$-hop controller)**
> We agree that some downstream tasks may require controllers of better properties. In that case, we can run the proposed algorihm to get a $\tau$-hop controller (which only learns the unstable component of the system). Then, we can run the $\tau$-hop controller to generate *stable* trajectories of the system and do system identification to estimate complete system matrices $(A,B)$ (there are many ways to do system identification from stable trajectories, e.g. the standard OLS method [Simchowitz2018, Rakhlin2018]). Then, based on the learned system, many other stable controllers can be designed based on the need of the downstream task. In this process, our algorithm plays an important role because it only lets the system blow up for $\tilde{O}(k)$ steps, as opposed to the $O(n)$ steps in the naive approach.

---

> > ### Author Response · Authors · 2022-08-02
> > **References**
> >
> > **References**
> >
> > [Rakhlin2018] Tuhin Sarkar and Alexander Rakhlin. Near optimal finite time identification of arbitrary linear dynamical systems, 2018, *arXiv: arXiv preprint 1812.01251*.
> >
> > [Simchowitz2018] Max Simchowitz, Horia Mania, Stephen Tu, Michael I. Jordan, and Benjamin Recht. Learning Without Mixing: Towards A Sharp Analysis of Linear System Identification, *Proceedings of the 31st Conference On Learning Theory (PMLR)*, 75:439-473, 2018.

---

> > ### Comment · Reviewer_ytQe · 2022-08-09
> > **Response**
> >
> > Thanks for the insightful comments regarding knowledge of $k$ and use of a $\tau$-hop controller.  I'm not fully convinced in the path to handling noise, but I think this is still a worth contribution and will keep my score as is.

---

### Official Review · Reviewer_v7qW · 2022-07-18

**Rating:** 6
**Confidence:** 3
**Soundness:** 3 good
**Presentation:** 3 good
**Contribution:** 3 good

**Summary:**

The paper deals with stabilization of an unknown LTI system from an trajectory of the system, thus learning and stabilization of the system must be done simultaneously. This is a timely problem and an current active research area. In the paper, the states/modes of the system are divided into stable/unstable, and only the unstable modes are identified and controlled.

**Questions:**

Can the weaknesses above (1 and 2) be addressed? By including a larger "k" in the algorithm compared to the number of unstable modes it may be possible to overcome or weaken these two limitations.

Can you state the theorems so that is clear what the assumptions are on the system parameters? Are all these parameters independent of the number of states?

**Limitations:**

Not applicable. This is basic science. I see no potential negative societal impact related to this work.

**Strengths And Weaknesses:**

Strengths: The paper well written, it deals with a timely problem, and as far as I know the main results are novel and improves on the state of the art.

Weakness:
1) There are several assumptions on the systems that restricts the applicability. Assumptions 4.1, 4.2, 4.3, seem rather harmless, however, the assumption on the eigenvalues |\lambda_1|^2 | |\lambda_k|<|\lambda_{k+1} restricts the applicability. Especially if |\lambda_1| is large.

2) The algorithm uses k, so one need to know the number of unstable modes.

3) In the key results, only dependence in k and n is stated but it is noted that the constant depends on other parameters as well. It seems to me difficult (and strange) to find a sequence of systems where all hidden parameters are constant (even for the ones listed). Here I think it is important to state exactly which parameters the assumptions on the parameters. In particular, when the number of states and unstable modes changes, the parameters need to be well defined and cannot, e.g., depend on the size of the system.

I did not read the supplementary material.

---

> ### Author Response · Authors · 2022-08-02
> **Response to Reviewer v7qW**
>
> Thank you for your insightful comments and suggestions!
>
> Our paper provides a new perspective on the learn-to-stabilize problem by providing instance-specific bounds that break the worst-case $2^{\Omega(n)}$  lower bound suggested by the literature. Our results rely on decomposing the state space into stable and unstable components and learning only the unstable component, which, to the best of knowledge, is a novel approach. We appreciate your comments on the limitation of the assumptions. As will be addressed point-by-point below, some of these assumptions can be easily relaxed based on the current techniques, while other assumptions require future work. We impose these assumptions because breaking the worst case $2^{\Omega(n)}$ lower bound is already a challenging problem, and hopefully our results can serve as an initial step and inspire future work that can push the boundary on the learn-to-stabilize problem.
>
> Please find below our point-by-point responses.
>
>
> **(On the assumption $|\lambda_1|^2 |\lambda_{k+1}| < |\lambda_k|$)**
> While this assumption seems strong, we point out that it is necessary for establishing theoretical bounds for our algorithm, and that in some special cases it could potentially be removed.
>
> The core reason why the assumption is needed is that the stable and unstable components are coupled with each other due to the non-orthogonality of $E_{\mathrm{u}}$ and $E_{\mathrm{s}}$, which makes the analysis of closed-loop stability challenging. For example, as can be shown in eq. (1) (on page 4, below line 166), any attempt by the controller to reduce the unstable component will inevitably increase the stable component, which then further propagates to the unstable component due to the off-diagonal block $\varDelta$ in eq. (1). To counter such side effects, we have to carefully design the $\tau$-hop controller to inhibit the coupling, and in doing so we do require certain assumptions on the eigen-gap.
>
> Relaxing this assumption is important and is left as future work. For example, based on our current techniques, this assumption can be relaxed when $\xi$ is sufficiently small (i.e., when $E_{\mathrm{u}}$ and $E_{\mathrm{s}}$ are almost orthogonal). In this setting, the off-diagonal block $\varDelta$ in (1) is close to $0$ and we don't need to use the $\tau$-hop controller. Hence we expect a cleaner bound (similar to that in the warm-up case) that does not depend on this eigen-gap assumption. Another potential way to relax the assumption is to avoid fully cancelling out the unstable component. Rather, we only have to find a stabilizing controller to *stabilize* the unstable component, which leaves some space to relax the assumption. All these are interesting directions and we leave them as future work.
>
>
> **(On the assumption that $k$ should be known beforehand)**
> This is not a limiting assumption because one can either (a) use a larger $k$ (as suggested by the reviewer) or (b) learn the actual instability index $k$. We elaborate on the two approaches below.
>
> (a) It is fine to use in the algorithm a $k$ larger than the true instability index — the algorithm still outputs a stabilizing controller because our analysis only relies on the ratio between eigenvalues rather than their values, and the complexity only suffers a little if $k$ is not set to be significantly larger.
>
> (b) It is possible to determine the true instability index by the singular value decomposition of $D$ (see line 233) as follows: after the initial $t_0$ steps, we form a matrix $D^{(i)} = [x_{t_0+1} ~ \cdots ~ x_{t_0+i}]$ when $x_{t_0+i}$ is obtained; then, we calculate the singular values of the $D^{(i)}$ matrix.  Whenever $D^{(i)}$ starts to contain some singular value that is almost $0$, we know $i-1$ is exactly the instability index. The reason why this is correct is that, when $i\leq k$, all singular values of $D^{(i)}$ should be larger than $1$; when $i>k$, since the system only has $k$ unstable modes, only $k$ singular values of $D^{(i)}$ will be larger than $1$, and the rest should be close to $0$.
>
>
> **(Instance-specific bounds)**
> A detailed table that summarizes the dependence on system parameters can be found at the end of Appendix G (page 36), and Section 4 (lines 308 through 330) provides some discussion on them. We agree that these parameters are *instance-specific*, and we use them to obtain the tightest bound for each system instance. Still, if an *instance-independent* bound is needed for a family of systems, where all parameters are uniformly bounded, we can simply replace those parameters with their bounds in eq. (41) through (46) (see Appendix G, pp. 35-36) and obtain an instance-independent bound.

---

### Meta-Review · Area_Chair_9MHb · 2022-08-24

**Recommendation:** Accept
**Confidence:** Less certain

**Metareview:**

The paper considers an interesting and timely theoretical problem in the intersection between control and machine learning. The paper is well-written and the results are novel and provide insight into how and when the sample complexity for stabilizing an unknown LTI system can be decreased. I thus recommend that it should be accepted. Nevertheless, I believe that the assumptions imposed are strong (full state measurements, strong controllability assumptions, requirements on the initial state, and systems with at least two unstable eigenvalues and no integrator dynamics) while the adaptive control literature in the 80's solved similar stabilization problems under much less stringent assumptions. Even though the contribution is theoretical, it would be useful with a physics-based example to make ideas and assumptions more concrete and compare the proposed approach against the state-of-the-art from both classical adaptive control and modern learning-based techniques.

**Award:**

No

---

### Decision · Program_Chairs · 2022-09-14

Accept